# Observing time-dependent energy level renormalisation in an ultrastrongly coupled open system

Alessandra Colla [1,2,4] ✉, Florian Hasse [1,4] ✉, Deviprasath Palani [1], Tobias Schaetz [1,3], Heinz-Peter Breuer [1,3] & Ulrich Warring [1] ✉

Understanding how strong coupling and memory effects influence energy levels in open quantum systems is a fundamental challenge. Here, we experimentally probe these effects in a two-level open system coupled to a single-mode quantum environment, using Ramsey interferometry in a trapped ion. Operating in the strong coupling regime, we observe both dissipative effects and time-dependent energy shifts of up to 15% of the bare system frequency, with the total system effectively isolated from external environments. These dynamic shifts, likely ubiquitous across quantum platforms, arise solely from ultra-strong system-mode interactions and correlation build-up and are accurately predicted by the minimal-dissipation Ansatz. Our approach identifies these as generalised Lamb shifts, matching conventional predictions on time-average. We provide experimental fingerprints supporting the Ansatz of minimal-dissipation, thereby suggesting it as a testable quantum thermodynamics framework and establishing a foundation for future benchmarks in strong-coupling quantum thermodynamics and related technologies.

Stable control and manipulation are critical for quantum technology and require a deep understanding of the underlying interactions of physical platforms with their surroundings. Open system theory has greatly advanced as a tool for predicting the evolution of a quantum system coupled to a general environment[1]. But more profound questions, e.g., how the energy of the system changes due to coupling to its environment are still lacking satisfactory answers, especially when the interaction is strong and memory effects are present.

For a macroscopic classical system that interacts with its surroundings, short-range interactions and scaling arguments justify neglecting the energy contribution associated with the system-environment interaction. However, for a quantum system, the interaction energy can be of comparable magnitude to that of the system. Thus, it can be expected to influence its energy and dynamics mainly when the environment is finite, structured, or considering strong couplings. Since, formally, the interaction energy is shared between the system and the environment, it is unclear how much of this energy locally affects the system, cp. Fig. 1. Consequently, predictions of open systems' energy levels are not uniquely defined, but play a crucial role in the design and control of quantum technology platforms[2–5]. Furthermore, determining the system's energy in strong coupling scenarios can provide insights on quantities like work and heat in non-equilibrium quantum thermodynamics[6,7], a field where existing theoretical approaches offer conflicting results[8–18].

A dynamically inspired out-of-equilibrium approach to this question predicts an effective energy-level renormalisation due to environmental interactions[19]. The effect generally depends on the parameters determining the total Hamiltonian, particularly the coupling strength, and details of the initial environmental states, such as its temperature. Known examples of the energy level renormalisation

[1]Institute of Physics, University of Freiburg, Hermann-Herder-Straße 3, D-79104 Freiburg, Germany. [2]Dipartimento di Fisica Aldo Pontremoli, Università degli Studi di Milano, Via Celoria 16, I-20133 Milan, Italy. [3]EUCOR Centre for Quantum Science and Quantum Computing, University of Freiburg, Hermann-Herder-Straße 3, D-79104 Freiburg, Germany. [4]These authors contributed equally: Alessandra Colla, Florian Hasse. ✉e-mail: alessandra.colla@unimi.it; florian.hasse@physik.uni-freiburg.de; ulrich.warring@physik.uni-freiburg.de

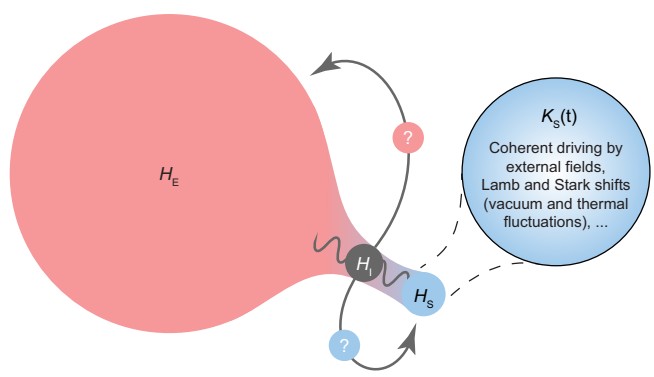

**Fig. 1 | Renormalisation of an open system Hamiltonian due to tracing out the environmental degrees of freedom.** In an open quantum system (Hamiltonian $H_S$), strong or structured interactions (Hamiltonian $H_I$) with some environment (Hamiltonian $H_E$) can lead to memory effects and enforce non-negligible energy contributions. In principle, the interaction affects both the system's reduced dynamics and energy. Note that the total system is assumed to be perfectly isolated from the rest of the universe: in an experimental setting, this means that a fully controllable bipartite system (system and environment) needs to be sufficiently isolated from the rest of the laboratory environment. We investigate an open-system approach to describe the renormalisation of the system energy levels when strongly interacting with a non-Markovian environment. The renormalisation is generally time-dependent and assumed to be uniquely encoded in the emergent Hamiltonian $K_S(t)$ via the Ansatz of minimal dissipation. Examples of known renormalisation in light-matter interaction derived from this approach are the Lamb and AC Stark shifts (for time-independent shifts) and semi-classical coherent driving (for time-dependent shifts). We validate this approach outside the known Markovian regimes by performing measurements on the open system only, providing strong evidence of the uniqueness of the effective system Hamiltonian in a non-Markovian scenario.

described by Colla and Breuer[19] for weakly coupled Markovian environments are given by the Lamb and AC Stark shifts induced in atomic energy levels by the vacuum and thermal fluctuations of the electromagnetic field[1,20,21]. These effects are captured by the Jaynes–Cummings (JC) model in the dispersive limit, where a two-level system is weakly coupled to a single bosonic mode with large detuning. They have been experimentally measured in a wide coupling range, e.g., from Rydberg atoms[22] to superconducting qubits[23] coupled to microwave cavities.

The JC model is indeed prototypical, as it describes not only physical situations in traditional cavity quantum electrodynamics, where a two-level atom is interacting with an optical or microwave cavity but also in several other modern experimental platforms – where the traditional constituents are replaced by analogue versions – such as quantum dots, superconducting qubits, and trapped ions[24]. For the purpose of this work, we highlight that the JC model also provides a fundamental playground to investigate the less understood strong coupling and non-Markovian regimes: although the model is exactly solvable via the dressed states' formalism, it can be cast into an open system problem by identifying the two-level system as the system (S) and the bosonic mode as the environment (E). Because of correlations and entanglement that build up between S and E, the dynamics of the two-level system includes dissipative contributions. It is also non-Markovian in general, due to the finite size of the mode as the environment and its associated recurrence effects. Moreover, certain experimental platforms, such as trapped ions and superconducting qubits, can probe stronger coupling regimes by moving closer to resonance and towards larger coupling strengths, while still maintaining near-perfect isolation from any kind of laboratory environment. This viewpoint has already been exploited in trapped-ion experiments[25,26], where the high degree of control over the entire JC model offers quality insights into strong coupling and non-Markovian

regimes. In such scenarios, the two-level system's energy levels may undergo renormalisation effects that are largely disregarded in theory and have yet to be investigated experimentally. According to[19], the predicted effective energy levels of the system could change with the duration of coupling, describing an effective driving of the energy levels of the open system even if the total Hamiltonian of the system-environment interaction is time independent. In quantum thermo-dynamics, this is interpreted as effective work, emerging from tracing out the degrees of freedom of the environment[27].

In this article, we investigate the dynamic effects of strong coupling and memory on the energy levels of open quantum systems by probing, in real time, the transition frequency of an open two-level system, experimentally realised with a single trapped $^{25}Mg^+$ ion. Operating near the ultra-strong coupling regime and close to resonance between the two-level system (S) and a single motional mode (quantum environment, E) initialised near the vacuum state, we observe pronounced dissipative effects on S due to its strong coupling with E. Our setup enables us to detect a time-dependent shift in the system's energy levels, reaching up to 15% of the bare spin frequency, in the absence of external driving. This time-dependent shift is a purely open-system effect, emerging from the two-level system's interaction with its environment, while the energy levels of the total system (described by the JC model) remain static, ensured by effective isolation from the laboratory environment. We interpret this type of shift as an emergent driving force on S, arising from correlation build up with E. The observed shift aligns accurately with predictions from the minimal-dissipation Ansatz. Time-averaged measurements and corresponding open-system predictions recover conventional Lamb shift and dressed-state energies, suggesting that this observed shift represents a generalised Lamb shift that is applicable across all coupling regimes and platforms. Our results provide strong evidence supporting the minimal-dissipation Ansatz[19], which stands out as a distinctly promising approach in quantum thermodynamics for making testable predictions, in particular, in the strong coupling regime. In this way, our findings offer unique insights into Hamiltonian renormalisation in open quantum systems, with potentially broad implications for quantum control platforms. Leveraging the trapped-ion platform, which can be scaled from simple to complex environments, our work establishes a solid foundation for observing, characterizing, and potentially harnessing time-dependent energy-level shifts, particularly in the context of strong-coupling quantum thermodynamics and technology applications.

## Results

### Minimal-dissipation Ansatz in the Jaynes–Cummings model

According to the minimal-dissipation Ansatz[19], the system's energy is described by an effective, potentially time-dependent Hamiltonian operator $K_S(t)$. This emergent Hamiltonian is a consequence of the joint dynamics of S and E and can be observed by probing solely S, see Fig. 1. We find $K_S(t)$ within the time-local master equation, which describes the exact evolution of S and is formally obtained with no approximations by tracing out the degrees of freedom of E. In very general terms, the master equation is written as a sum of two terms, namely a commutator with the Hamiltonian $K_S(t)$ and a dissipator term[28–30]

$$
\begin{aligned}
\frac{d}{dt}\rho_S(t) = &-i\big[K_S(t), \rho_S(t)\big] \\
&+ \sum_k \gamma_k(t)\Big[L_k(t)\rho_S(t)L_k^\dagger(t) - \frac{1}{2}\big\{L_k^\dagger(t)L_k(t), \rho_S\big\}\Big],
\end{aligned}
\tag{1}
$$

where the time-dependent rates $\gamma_k \in \mathbb{R}$ can be negative. A master equation in this form (also sometimes referred to as *generalised Lindblad form*) axiomatically exists for all open system evolutions, except for at most a set of discrete points on the time axis[30,31], even if S

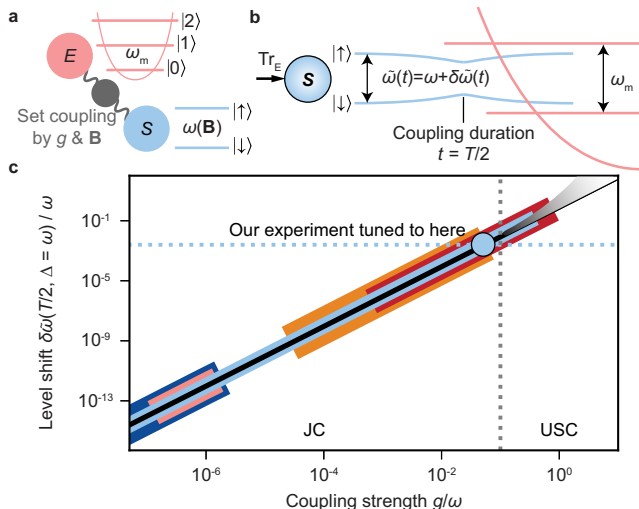

**Fig. 2 | Perspective on paradigmatic open-system dynamics, validating the time-dependence of significant energy renormalisation in the Jaynes–Cummings (JC) model near the ultrastrong coupling (USC) threshold.** **a** Experimentally, we utilise a single bosonic mode (cavity analogue) with eigenfrequency $\omega_m$ of a trapped magnesium cation as environment $E$ and, as system $S$, the dressed electronic states (atom analogue) ($|\uparrow\rangle$, $|\downarrow\rangle$) with bare spin frequency $\omega(\mathbf{B})$ tuned by a static effective magnetic field $\mathbf{B}$ (Zeeman-shift analogue). We can continuously tune the $S$-$E$ coupling strength $g$ near the USC threshold in the red sideband regime[45]. **b** When tracing out the environment's degrees of freedom, the Ansatz of minimal dissipation[19] predicts a significant energy shift as a function of coupling duration $t$ with a maximum level shift $\delta\tilde{\omega}(T/2) = -\frac{2g^2}{\Delta}$ for the mode initially in the motional ground state; see Supplementary Information. This effect can be resonantly enhanced for small detuning $\Delta(\mathbf{B}) = \omega_m - \omega(\mathbf{B})$. **c** The shift is estimated in the JC model to be $\propto g^2$ and is shown as a function of $g$ (black line) for several orders of magnitude. We emphasise the significance of different experimental platforms[24]: Atoms in optical (dark blue) and microwave (light red) cavities, quantum dots (orange), superconducting qubits (red) and trapped atomic ions (light blue). The target regime of our work (light blue disk) is near the USC, an atypically strong coupling regime for trapped-ion experiments, since the coupling strengths are typically orders of magnitude lower. Beyond the USC, the JC approximation loses its validity toward the deep strong coupling regime.

and $E$ are initially correlated[32]. It is also already known exactly for a set of integrable models, e.g., for systems linearly coupled to bosonic environments[33–36], various dephasing models[37] and the JC model[38]. When no exact treatment is available, the master equation can still be evaluated numerically or approximated through a time-convolutionless perturbation expansion[39–41], describing all effects from strong coupling and non-Markovianity, and that can still be put in the form of Eq. (1).

The splitting between a commutator and a dissipator in the master equation, and thus the exact expression for the Hamiltonian $K_S$, is, however, highly non-unique[1,42]. Nonetheless, given any open system evolution, the splitting can be uniquely assigned by imposing a recent minimal-dissipation Ansatz[43] that requires minimisation of the dissipator's action as a superoperator by averaging over input and output states to recognise the part of the evolution which is coherent and retrievable by $S$. This is taken in ref. 19 to be an additional physical principle that uniquely specifies the emergent Hamiltonian $K_S(t)$ and, consequently, the system's modified energy levels. These energy levels reproduce the known Lamb and AC Stark shifts[1] in the appropriate cases, as well as coherent driving in a semi-classical limit, where a time-dependent Hamiltonian emerges from the coupling with coherent modes while dissipative contributions are negligible[44]. Here, we investigate whether the minimal-dissipation Ansatz reproduces observable energy shifts also in other regimes, particularly under strong coupling, finite environments, or resonant conditions.

As depicted in Fig. 2a, in the following, we consider $S$ given by a two-level system, which is coupled to a single bosonic degree of freedom acting as $E$, described by the JC Hamiltonian

$$H = \frac{\hbar\omega}{2}\sigma_z + \hbar\omega_m a^\dagger a + \hbar g(\sigma_+ a + \sigma_- a^\dagger), \qquad (2)$$

where $\sigma_z$ is the Pauli $z$-operator, $\sigma_+$ and $\sigma_-$ and $a^\dagger$ and $a$ are the raising and lowering operators of the spin and the mode, respectively. The bare spin and mode frequencies are given by $\omega$ and $\omega_m$, respectively, while $g$ denotes the coupling parameter, and $\hbar$ is the reduced Planck constant.

The model can be studied considering the two-level system from an exact open-system perspective, where the environmental degrees of freedom of the mode are traced out without making any further approximations on coupling strength or time scales. The evolution of the density matrix of $S$ is thus described by a completely positive and trace preserving dynamical map and by a time-local master equation[1]. For an initial thermal state of the mode, described by mean excitation number $\bar{n} = (e^{\beta\hbar\omega_m} - 1)^{-1}$ with $\beta$ its initial inverse temperature, the exact time-convolutionless master equation for $S$, in generalised Lindblad form of Eq. (1) and in minimal dissipation form, reads[38]:

$$\dot{\rho}_S = -i\left[\frac{\tilde{\omega}(t)}{2}\sigma_z, \rho_S\right] + \gamma_z(t)\left[\sigma_z\rho_S\sigma_z - \rho_S\right]$$
$$+ \gamma_+(t)\left[\sigma_+\rho_S\sigma_- - \frac{1}{2}\{\sigma_-\sigma_+, \rho_S\}\right] \qquad (3)$$
$$+ \gamma_-(t)\left[\sigma_-\rho_S\sigma_+ - \frac{1}{2}\{\sigma_+\sigma_-, \rho_S\}\right]$$

with time-dependent coefficients $\tilde{\omega}, \gamma_{+,-,z}$ depending on $\bar{n}$ and on the Hamiltonian parameters, $g$ and the spin-mode detuning $\Delta = \omega_m - \omega$. The master equation, Eq. (3), reveals an emergent Hamiltonian of the form $K_S(t) = \tilde{\omega}(t)\sigma_z/2$, where the transition frequency between ground and excited state is renormalised via a time-dependent shift $\delta\tilde{\omega}(t)$, i.e., $\tilde{\omega}(t) = \omega + \delta\tilde{\omega}(t)$. The dependence on the coupling duration is due to the mode environment's finite size and the related presence of memory effects. It can be interpreted as an energy exchange between the system and the environment, which appears as a driving of the system, as schematically illustrated in Fig. 2b. The renormalisation effect on the spin degree of freedom is general and will be of the form shown above for any platform that can be described by the JC model. Because the average magnitude of the renormalisation is proportional to the square of $g$ (see Supplementary Information), the effect will have varying impact on each experimental platform depending on its natural scale of coupling strength. Usually, the energy level shift remains several orders of magnitude smaller than the bare spin energy splitting; see Fig. 2c. But trapped ions and superconducting qubits can tap into the ultrastrong coupling (USC) regime[24], where the renormalisation contribution reaches about 10–20% of $\omega$.

The transition frequency of $S$, and thus its (possibly time-dependent) energy level splitting, can be detected via its Larmor frequency. In our case, it holds analytically that the Larmor frequency is identical to the renormalised energy splitting in the Hamiltonian $K_S(t)$[38]. While this is true for any initial temperature of the mode, the special case of the mode initially in the vacuum gives an analytic, explicit expression for $\delta\tilde{\omega}(t)$, which is periodic in time with the $S$-$E$ coupling duration $T(\Delta) = 2\pi/\sqrt{\Delta^2 + 4g^2}$, and reads

$$\delta\tilde{\omega}(t) = -\frac{2g^2}{\Delta}\frac{1}{1 + \left(1 + \frac{4g^2}{\Delta^2}\right)\cot^2\left(\frac{\pi}{T(\Delta)}t\right)}. \qquad (4)$$

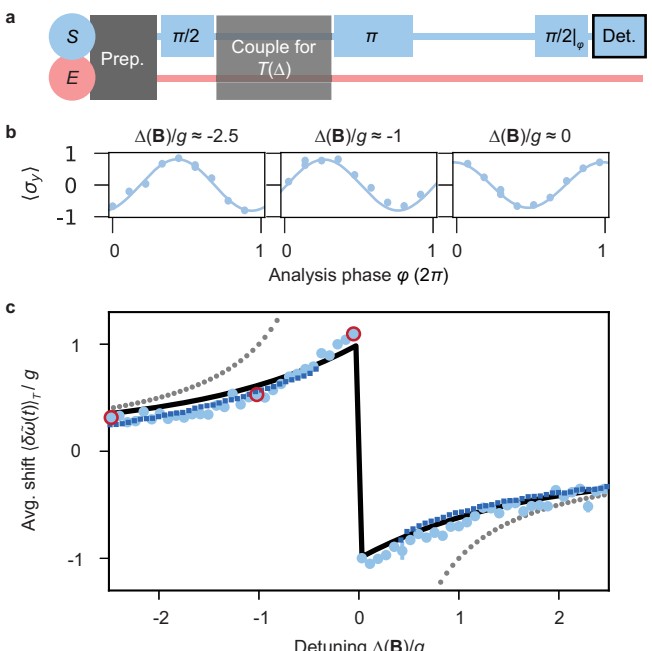

**Fig. 3 | Experimental probing of time-averaged energy renormalisation for narrow detuning range.** The experimental sequence is shown in **a**: The motional mode ($E$) with frequency $\omega_m = 2\pi\,1.303(1)$ MHz is cooled close to the motional ground state with a residual thermal contribution of $\bar{n} = 0.08(3)$. A first $\pi/2$-pulse initialises the spin ($S$) in a superposition state with equal weights. The $S$-$E$ coupling pulse with a coupling strength of $g \simeq 2\pi\,0.065$ MHz is applied for a duration of $T(\Delta)$, followed by a spin flip ($\pi$-pulse) and a free evolution matching $T(\Delta)$. A second $\pi/2$-pulse with variable analysis phase $\varphi$ is applied, succeeded by fluorescence detection. **b** Raw data (light blue disks) with sinusoidal fit (light blue solid line) for three distinct detunings $\Delta(\mathbf{B})/g$. Every data point consists of 100 experimental repetitions of the same parameter settings. The phase change is proportional to the time-average shift. A detailed description of the analysis used is given in the Methods section. Error bars are smaller than the data points. **c** Following the analysis procedure, we show experimental results (light blue disks); red circles mark results from the selected raw data from **b**. For $\Delta > 0$ the data points match the JC prediction (Eq. (5); black solid line) and TI simulation (dark blue dotted line), as well as, for $\Delta \gg g$ agreement with the Lamb-shift (grey dotted line). All error bars represent the standard error of the mean.

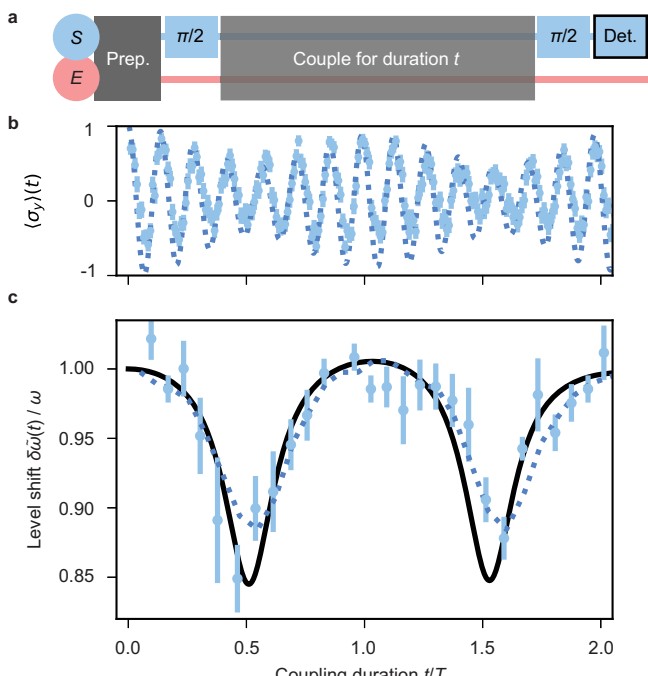

**Fig. 4 | Observing time-resolved Larmor frequency modulation as a proof of time-dependent energy renormalisation near the USC regime. a** In our experiment, we initialise the spin ($S$) in equal superposition of $|\uparrow\rangle$ and $|\downarrow\rangle$, and the mode ($E$) with a frequency of $\omega_m = 2\pi\,1.304(1)$ MHz near the vacuum state with a residual thermal contribution of $\bar{n} = 0.08(2)$. We realise $\omega = 2\pi\,1.24(3)$ MHz (free Larmor frequency), and coherently couple the spin to the mode for variable duration with the coupling strength $g \simeq 2\pi\,0.078(2)$ MHz, resulting in a fixed detuning $\Delta(\mathbf{B})/g \simeq 0.8$. **b** In interleaved experimental sequences, we determine expectation values of $\langle\sigma_y\rangle(t)$ as a function of coupling duration $t$. We show 242 data points (light blue disks), 600 experimental repetitions per data point, and the corresponding TI simulation (dark blue dotted line). We attribute the periodic loss and recovery of contrast to dissipation (build up of correlations) of the system into the environment; due to the finite size of the environment, we observe high-contrast revivals, characteristic of non-Markovian effects. **c** Following this procedure, we determine the normalised Larmor frequency, $\delta\tilde{\omega}(t)/\omega$, as a function of coupling duration from six runs and show averages as light blue disks. The black solid line represents the analytical prediction of energy renormalisation in the JC regime. In contrast, the dark blue dotted line results from TI simulations. We observe a maximal shift of $\simeq 15\%$ of the bare spin frequency. All error bars represent the standard error of the mean.

In this zero-temperature case, the time-average shift is given by

$$\langle\delta\tilde{\omega}(t)\rangle_{T(\Delta)} = \frac{-2g^2\,\mathrm{sgn}(\Delta)}{|\Delta| + \frac{2\pi}{T(\Delta)}}, \tag{5}$$

and it is identical to the dressed-state energy of the total Hamiltonian, Eq. (2), associated with the excited spin; see Supplementary Information. We remark that the dressed energies appear only as time-averages of the spin frequency shift, which is, however, from the open system perspective, fundamentally time-dependent even when the mode is initially in the vacuum.

## Experimental evidence for time-averaged shifts

We implement the model experimentally with a single trapped magnesium cation, choosing $E$ represented by a single motional mode with $\omega_m \simeq 2\pi\,1.3$ MHz, while the energy of $S$ is tuned by an effective static magnetic field $\mathbf{B}$ (analogue to a Zeeman shift), near $\omega(\mathbf{B}) \approx \omega_m$, cp. Fig. 2a. We ensure near identical dynamics as the JC model and perform the experiments in the so-called red sideband regime[45], see Supplementary Table 1. Further, we perform numerical simulations of the trapped-ion (TI) Hamiltonian, neglecting technical limitations[26,46,47], see Supplementary Fig. 2. In-depth information about the experimental setup, data recording, and analysis are given in Methods. In contrast,

numerical simulations and comparisons of the JC approximation to the red sideband regime are shown in Supplementary Figs. 1 and 2.

We employ Ramsey interferometry to measure the frequency of $S$ and observe its shift due to the described renormalisation effect. We first assess the average shift $\langle\delta\tilde{\omega}(t)\rangle_T$ as a function of $\Delta$ and then resolve the time-dependent frequency shift $\delta\tilde{\omega}(t)$ in a close to resonance case. Both sequences are depicted and explained in Figs. 3a and 4a, respectively.

In the first sequence, to probe $\langle\delta\tilde{\omega}(t)\rangle_T$, we couple $S$ and $E$ for duration $T(\Delta)$. The sinusoidal model fits the raw data, as illustrated in Fig. 3b, and yields the accumulated phase of coherences in $S$ and, thus, the average energy shift after $T(\Delta)$. Figure 3c displays the probed $\langle\delta\tilde{\omega}(t)\rangle_T$ for a small detuning range, including the JC model prediction, Eq. (5), the TI simulation and the Lamb shift. In the dispersive limit $|\Delta| \gg g$, the Lamb shift agrees with $\langle\delta\tilde{\omega}(t, |\Delta|\gg g)\rangle_T$ and is given by $-g^2/\Delta$[22,23]. We find agreement between experimental and numerical data (only performed for $|\Delta|/g > 0.25$), as well as convergence to the Lamb-shift prediction. In addition, for $\Delta > 0$, the minimal-dissipation Ansatz, Eq. (5) agrees on a similar level, while for, $\Delta < 0$, we find only qualitative agreement. We attribute these systematic deviations to residual thermal contributions and the impact of a carrier term in the

TI Hamiltonian; see Supplementary Information. Nonetheless, this demonstrates the power of the open-system perspective and the Ramsey sequence (traditionally used exclusively in the dispersive regime) to recover dressed-state energies near resonance by a systematic phase accumulation encoded in $S$.

### Experimental evidence for a generalised (time-dependent) Lamb-shift

In our second sequence, to resolve a time-dependent variation, we choose $\Delta(\mathbf{B})/g \simeq 0.8$ and vary the $S$-$E$ coupling duration $t$. As shown in Fig. 4b, we provide raw data of six accumulated measurement runs (light blue data points) of the expectation value of $\sigma_y(t)$. We further depict numerical TI simulations of the experimental sequence (dark blue dotted line), which agrees with the raw data. We observe varying amplitudes indicating dissipative effects on $S$ due to correlations with $E$ (the motional mode). The amplitudes show alternating decoherence and recoherence which result from strong memory effects, in particular recurrences, due to the finite size of $E$. We evaluate the instantaneous Larmor frequency by counting the zero-crossings of $\sigma_y(t)$; this simplified procedure allows us to automatically average out fast oscillating contributions that are due to the counter-rotating terms in the TI Hamiltonian and to keep only the leading renormalisation effect due to the JC interaction, cf. Methods. The measured energy splitting, depicted in Fig. 4c, shows a modulation with maximum variations of about 15% and an average shift of $\simeq 4\%$ of $\omega$. The data points in Fig. 4c describing the time-dependent energy shift of $S$ indeed agree with the theoretical prediction (black solid line in Fig. 4c) made using the method of minimal dissipation[19] applied to the JC model, cp. Eq. (4), and provide direct evidence of time-dependent renormalisation of the spin energy levels. Furthermore, we observe a slight asymmetry in the height of the two recorded dips due to non-ideal vacuum-state preparation. The average motional expectation value $\bar{n} = 0.08(2)$ corresponds to a low but finite mode temperature and impacts the energy levels, which would be otherwise periodic at zero temperature. The influence of the initial environmental temperature is also an effect that is not directly appreciable at the level of total dressed energies, but is predicted by the minimal-dissipation Ansatz. More dramatic changes could, in principle, be observed at higher temperatures; in such a parameter regime, however, the JC Hamiltonian, cp. Eq. (2), is no longer suitable for the prediction of the TI evolution[45,46].

## Discussion

The time-dependent energy shift recorded here can be considered a generalisation of the well-known time-independent Lamb shift. It can be regarded as resulting from an energy exchange between $S$ and $E$, which manifests itself as time-dependent driving terms in the effective Hamiltonian, even though the total Hamiltonian is time-independent. We have carried out the experiment on a purposefully simple model (the JC model), which in our platform is realised as a fully controllable closed quantum system due to the negligible coupling to the rest of the laboratory environment. We observed the open quantum system (the two-level system) strongly coupled to the non-Markovian environment (the mode), by expanding the well-known Ramsey sequence to resolve time-dependent energy shifts in the system. This procedure is general and applicable to all quantum platforms studying qubits, encouraging further experiments on time-dependent energy level renormalisation both within and beyond the JC model. The variation in spin energy levels, like amplitude variations, reflects an open-system effect tied to the build up of $S$-$E$ correlations. These findings suggest that Lamb and AC Stark shifts arise from such correlations and their temporal evolution. We propose that these ubiquitous light shifts fundamentally stem from time-dependent effects that are typically averaged out and observed only in far-detuned limits. Our study also suggests that the emergent Hamiltonian $K_S$ (following the Ansatz of minimal dissipation), which generally exists for any open system, is promising in predicting

energy renormalisation in other physical systems. Since $K_S$ depends entirely on the open system dynamics, it can, in principle, be retrieved experimentally via process tomography[48], generally accessible by any quantum technology platform. Further, using the versatile trapped-ion platform, explorations focus on understanding the emergent Hamiltonian for the analogue spin degree of freedom at higher mode temperatures and investigating the effects of non-thermal environmental states, such as coherent, squeezed, or Fock states. By scaling the trapped-ion platform from single to multimode environments with diverse initial states, e.g., vacuum, thermal, displaced, or entangled multimode squeezed states, the approach enables gradual increases in complexity, potentially uncovering subtle energy shifts. The emergent driving appearing in these situations could then be exploited for the understanding of the thermodynamic properties of quantum systems, and it allows for testing the prospects of development of autonomous quantum heat engines[49], with the possibility of achieving enhanced efficiencies in non-Markovian scenarios[27]. Outside the current platform, the following steps include studying the renormalisation of multiple energy levels and the possible observation of driving due to a non-Markovian but continuous environment. Finally, the time-dependent Lamb-shift type renormalisation observed here opens a direction for future exploration, which aims at a consistent formulation of open system dynamics within the framework of quantum field theory, connecting concepts from renormalisation theory with effective dynamical laws obtained from an open system approach[1]. Such a formulation will facilitate a more general and more profound analysis of the renormalisation of parameters in reduced master equations, such as our Eq. (3), leading to insights into nonequilibrium quantum thermodynamics through a field-theoretical perspective.

## Methods

### Experimental setup

Our experimental platform represents an overall nearly perfectly isolated quantum system on all relevant time scales[26,46,47], from which we select two appropriate subsystems $S$ and $E$: we choose a single $^{25}\text{Mg}^+$ ion trapped in a linear Paul trap, with a drive frequency of $\simeq 2\pi\,56.3$ MHz. Due to the nuclear spin of $^{25}\text{Mg}$ of 5/2, the applied homogeneous magnetic quantisation field $|\mathbf{B}_{\text{quant}}| \simeq 0.58$ mT induces a hyperfine splitting. We define as pseudo spin (qubit) degree of freedom (DOF) $|F=3, m_F=3\rangle = |\downarrow\rangle$ and $|F=2, m_F=2\rangle = |\uparrow\rangle$, where $F$ is the total angular momentum and $m_F$ is the projection of the angular momentum along the magnetic field axis, resulting in a transition frequency $\omega^* \simeq 2\pi\,1.8$ GHz. The phonon DOF is described by three decoupled harmonic oscillators with frequencies $2\pi$ {1.3, 2.9, 4.5} MHz. The lowest mode is oriented approximately along the axial direction, while the two higher modes are oriented in the radial plane. The cycling transition is tuned near the $S_{1/2}$-to-$P_{3/2}$ transition ($\simeq 280$ nm) and is used for Doppler cooling and detection, the regarding $\mathbf{k}_c$-vector of the laser beams are aligned along $\mathbf{B}_{\text{quant}}$. Additional laser beams are used for repumping and state preparation to couple to appropriate Zeeman substates of $S_{1/2}$ and $P_{1/2}$. To determine the electronic state population, the fluorescence of the ion is detected via a photomultiplier tube. For further evaluation, we analyse the photon histograms[45]. For determining the thermal contribution $\bar{n}$, the population distributions of the motional states are reconstructed by mapping them onto the electronic states[45]. For this, we utilise a two-photon stimulated-Raman (TPSR) transitions, which are detuned from the $S_{1/2}$-to-$P_{3/2}$ transition by $\simeq + 2\pi\,20$ GHz, and the effective $\mathbf{k}_R$-vector of the used TPSR beam combination coincide with the axial eigenmode orientation enabling carrier and sideband transitions with tunable Rabi rates $\Omega_R$ between $2\pi$ {0.05, 0.6} MHz and fixed Lamb-Dicke parameter $\eta \simeq 0.4$. Tuning the TPSR beams into the red sideband regime yields a Hamiltonian that is, in a first-order approximation, formally equivalent to the JC Hamiltonian. The axial eigenmode ($E$) and laser-dressed electronic states ($S$) are coupled via the TPSR beams, cp. Table 1 and Supplementary Information.

**Table 1 | Comparison between the Jaynes–Cummings and trapped-ion (near the red sideband regime) models**

| | Jaynes–Cummings model | Trapped-Ion model (red sideband regime) |
|---|---|---|
| Composition of $S$ and $E$ with characteristic (bare) frequencies $\omega$ and $\omega_m$ | Two-level atom and a single mode of the quantized electromagnetic field (photons) | Laser-dressed states of ion's internal levels and quantized vibrational modes (phonons) |
| Coupling strength | Direct $g$ | Laser mediated $\eta\Omega_R/2$ |
| Detuning $\Delta = \omega_m - \omega$ | Depends on the static effective magnetic field **B**, leading to a detuning $\Delta(\mathbf{B})$ | Set by the frequency difference to the first red sideband, $\Delta_R$ |

More details on this in ref. 45 and see Supplementary Information.

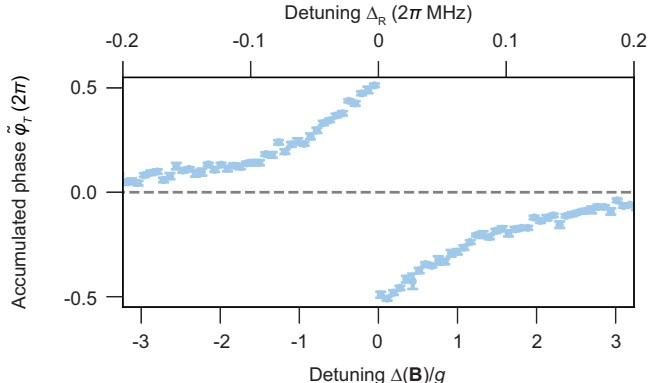

**Fig. 5 | Accumulated phase $\varphi_T$ for various detunings $\Delta(\mathbf{B})/g$.** We show the results of sinusoidal model fits of the raw data taken to determine $\langle\delta\tilde{\omega}(t)\rangle_T$, contributing to Fig. 3c. Error bars represent the standard deviation of the mean.

In our setup, we can continuously tune the coupling strength $g/\omega \simeq \{10^{-2}, 10^{-1}\}$ and we choose for the presented experimental results $g/\omega \simeq 0.05$, still below, but close to, the USC regime, cf. Fig. 2c.

### Determination of time-average shift

The initial step to determine $\langle\delta\tilde{\omega}(t)\rangle_T$ is fitting a negative cosine function $-C/2\cos(\varphi + \tilde{\varphi}_T) + 0.5$ with the two free parameters contrast $C$ and accumulated phase $\tilde{\varphi}_T$ to the histogram analysed raw data. This is exemplarily depicted in Fig. 3b, for three selected $\Delta(\mathbf{B})/g$, with the boundary condition $\tilde{\varphi}_T \xrightarrow{|\Delta|\gg 0} 0$. This boundary condition arises from the fact that $T(\Delta) \xrightarrow{|\Delta|\gg 0} 0$, hence, the two arms of the Ramsey interferometer are identical, resulting in no expectable phase accumulation. Note that additional waiting durations are implemented between the $\pi/2$ and the $\pi$ pulses, which ensures that, independent of the coupling pulse duration $T(\Delta)$, both arms have an equal duration for all $\Delta$. The measured accumulated phase, illustrated in Fig. 5 is related to $\langle\delta\tilde{\omega}(t)\rangle_T$ via $\tilde{\varphi}_T = \int_0^T \tilde{\omega}(t)\,dt = \langle\tilde{\omega}(t)\rangle_T \cdot T$ with $\tilde{\omega}(t) = \omega + \delta\tilde{\omega}(t)$. Following these relations and assuming $\langle\tilde{\omega}(t)\rangle_T$ to be static, to obtain $\langle\delta\tilde{\omega}(t)\rangle_T$, we divide $\tilde{\varphi}_T$ by $T(\Delta)$ and receive the result shown in Fig. 3c.

### Determination of time-dependent shift/Larmor frequency

In our analysis, we use a customised algorithm to detect the frequency of zero crossings in the time series signal of $\langle\sigma_y\rangle(t)$, as shown in Fig. 4b. Utilising our knowledge of $\omega$, the algorithm clusters adjacent crossings to reduce the impact of fast oscillations or noise. We identify the median times, $T_{\mathrm{zero},i}$ (where $i$ is the number of zero crossings), within these clusters to determine the zero-crossing timings, where $\langle\sigma_y\rangle(T_{\mathrm{zero},i}) \simeq 0$. Finally, the time-dependent Larmor frequency values, $\omega_L(t)$, (Fig. 4c) are estimated from the timing differences between neighbouring zero crossings using the formula: $\omega_L(t_i) = \{\pi/(T_{\mathrm{zero},i+1} - T_{\mathrm{zero},i})\}$.

### Data availability

All the data used to generate figures in this paper are provided in the Source Data file. Source data are provided with this paper.

## Code availability

Simulation codes used in this study are provided in the Source Data file.

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

## Acknowledgements

We thank Frederike Doerr for her help checking data evaluation and making comparative measurements. A.C. would like to thank Edoardo Carnio and Janine Franz for their precious insights. This project has received funding from the European Union's Framework Programme for Research and Innovation Horizon 2020 (2014-2020, H.P.B.) under the Marie Skłodowska-Curie Grant Agreement No. 847471 (A.C.), the Deutsche Forschungsgemeinschaft (DFG) (Grant No. 217 SCHA 973/6-2, T.S.), and the Georg H. Endress Foundation (H.P.B and T.S.). This publication was supported by funding from the Projekt DEAL agreement (https://deal-konsortium.de/ueber-deal), enabling open-access publication.

## Author contributions

A.C. developed the theory under the supervision of H.P.B. F.H. performed the experiment and data analysis under the supervision of T.S. and U.W. D.P. assisted in preparing the experimental setup and executing the experiment. U.W. performed numerical trapped-ion and Jaynes–Cummings simulations. A.C. and F.H. contributed equally to this work and wrote the paper with all authors' input.

## Funding

## Competing interests

The authors declare no competing interests.
