## [Transparent Peer Review file · Nature Communications]

Observing Time-Dependent Energy Level Renormalisation in an Ultrastrongly Coupled Open System

Corresponding Author: Mr Florian Hasse

Version 0:

Reviewer comments:

Reviewer #1

(Remarks to the Author)

The manuscript describes an experiment that studies the shift of energy levels of a two level system coupled to a harmonic oscillator. Authors implemented this coupling experimentally by driving a motional sideband transition of a trapped ion and measured the additional phase acquired by the two level system during the interaction time. Based on this measurement authors determine the effective change of energy levels of a 2 level system due to coupling with the environment that consists of a single harmonic oscillator. The experiment was done in the strong coupling regime where the Rabi frequency of the sideband transition is comparable to the energy splitting of the 2 level system. The experimental techniques described in the manuscript are quite standard and all were employed before in many trapped ion experiments.

The main motivation of the experiment is the test of the theory presented in Ref. 19. However, it looks like the experimental system that authors employ is too simple for this purpose and the results can be explained well by simpler and well established theories. The physical system of a qubit interacting with a harmonic oscillator realized in experiment is a closed system with no significant dissipation. Initial state of this system is close to a pure state because the initial state of the motional mode is close to the vacuum state. In this case, a traditional dressed state approach looks sufficient to explain the shift of the energy levels, as the authors themselves pointed out at the end of the supplemental information, and the concepts of "open system ansatz of minimal dissipation" or "direct energy renormalization" are redundant and unnecessary.

In my opinion, authors failed to establish a clear connection between the theoretical hypothesis they intended to test and the experiment they have actually performed. Therefore I cannot recommend publication of the manuscript in the present form.

Reviewer #2

(Remarks to the Author)

The authors address the complex and challenging problem of understanding how strong coupling and memory effects influence the energy levels of open quantum systems. In their work, they demonstrate these effects by probing the transition frequency of an open two-level system within the Jaynes-Cummings model, experimentally realized through Ramsey interferometry in a single trapped $^{25}\text{Mg}^{+}$ ion. Their measurements, conducted while the system is coupled to a single-mode environment, reveal a time-dependent shift in the system's energy levels, reaching up to 15% of the bare system frequency. This shift, accurately predicted using an open system approach with minimal dissipation, arises solely from ultra-strong system-mode interactions and the buildup of correlations. Time-averaged measurements align with the dispersive Lamb shift predictions and match dressed-state energies, indicating that the observed shift represents a generalized Lamb shift applicable across all coupling and detuning regimes. The authors provide direct evidence of dynamic energy level renormalization in strongly coupled open quantum systems, even though the total system-environment Hamiltonian remains static. This underscores the critical role of memory effects in shaping the energy landscape of the reduced system. Their results offer deeper insights into Hamiltonian renormalization, which is vital for strong-coupling quantum thermodynamics and essential for advancements across all quantum platforms.

The manuscript is well written and very clearly presented. The predictions are strongly supported by the experimental data, which are of quite high quality. The paper will definitely be interesting to a ample range of scholars, including those working in the field of finite-time quantum thermodynamics and strong-coupling quantum thermodynamics.

I support the manuscript for publication in light of the exemplary clarity and quality of the results being reported.

Version 1:

Reviewer comments:

Reviewer #1

(Remarks to the Author)

In the revised manuscript authors have clarified connection between predictions of the minimal dissipation ansatz and experimental measurements they have performed. While it still does not look to me that minimal dissipation ansatz is a unique way to interpret presented experimental data, the idea of the authors is clear now. As far as I am concerned the revised version of the manuscript is suitable for publication.

Reviewer #2

(Remarks to the Author)

Thank you for contacting me about the revision of this manuscript. I found the revised version of the work and the response provided by the authors compelling and quite convincing, even in terms of the quality of the authors' replies to some of the comments raised by Reviewer#1.

I do not have any concerns on the suitability of the manuscript for publication in Nature Communications and, indeed, would like to strongly recommend that the work is accepted.

Thank you for the opportunity to revise our manuscript, “*Observing Time-Dependent Energy Level Renormalisation in an Ultrastrongly Coupled Open System.*” We are grateful for the constructive feedback from the reviewers, which has helped us clarify our findings and enhance the manuscript. Below, we summarize the key points of our response and the revisions made:

1. **Significance of Findings:** Reviewer 2 acknowledges that our work provides “*direct evidence of dynamic energy level renormalisation in strongly coupled open quantum systems, even though the total system-environment Hamiltonian remains static.*” This highlights the experimental precision and the importance of our findings in addressing unresolved questions in quantum thermodynamics.
2. **Contrasting Perspectives:** The contrasting opinions of the reviewers reflect an active debate on frameworks for understanding strong coupling and memory effects. Reviewer 2 emphasises that our predictions are strongly supported by high-quality experimental data, while Reviewer 1’s concerns have been directly addressed through additional clarifications and expanded discussion.
3. **Advancing the Field:** Our study validates the minimal dissipation Ansatz, a debated yet promising framework for predicting time-dependent energy shifts – phenomena inaccessible to traditional methods. By operating in a strong coupling regime, atypical for trapped ions, and employing state-of-the-art Ramsey sequences, we provide experimental evidence for effects long considered contentious. We note that our experimental platform can be extended in size and dimension, ion by ion, to further test the validity of the Ansatz in increasingly complex settings.
4. **Broader Relevance:** Our results have implications across quantum platforms, including superconducting qubits and structured baths, and establish a foundation for future exploration of time-dependent renormalisation in complex environments.

We believe the revisions and clarifications made in the manuscript strengthen its impact, and, to the best of our knowledge, we address all concerns raised by the reviewers point by point below. All changes are shown in the manuscript text file with colour highlighting.

Point-to-point response to Reviewer #1:

Reviewer: *The manuscript describes an experiment that studies the shift of energy levels of a two level system coupled to a harmonic oscillator. Authors implemented this coupling experimentally by driving a motional sideband transition of a trapped ion and measured the additional phase acquired by the two level system during the interaction time. Based on this measurement authors determine the effective change of energy levels of a 2 level system due to coupling with the environment that consists of a single harmonic oscillator. The experiment was done in the strong coupling regime where the Rabi frequency of the sideband transition is comparable to the energy splitting of the 2 level system. The experimental techniques described in the manuscript are quite standard and all were employed before in many trapped ion experiments.*

Our Response: We thank the reviewer for their summary, which captures the main points of our study. Yet, we’d like to clarify two key aspects to underscore the novelty and broader implications of our work.

1. Our experiment operates in a strong coupling regime, which is typically unreachable in trapped-ion setups. This is partially to maintain Lamb-Dicke parameters far below one for high-fidelity quantum information processing. However, the strong coupling regime is prerequisite to study open-system effects under fully controlled conditions, where the total system (in our case, a single spin and a single mode) is near perfectly closed. By deliberately entering this regime – where the Rabi frequency of the sideband transition is comparable to the two-level system’s energy splitting – we can detect related effects

at a sufficiently high signal-to-noise ratio. This unique parameter space allows probing the relevant dynamic and dissipative effects that remain unexplored in standard settings. Furthermore, this regime is also relevant to other quantum platforms, such as superconducting qubits, thereby broadening the impact and applicability of our findings across the field.

2. While we utilize established experimental techniques, we advance state-of-the-art methods to measure light-like shifts and achieve the direct observation of a time-dependent Lamb-like shift. To the best of our knowledge, this is a first-time observation. Our result validates the minimal dissipation Ansatz, a novel and debated approach within the open-system framework that uniquely predicts time-dependent shifts in system energy – phenomena beyond the reach of conventional theoretical methods. By demonstrating the practical relevance of this framework, we see, our work addressing unresolved questions in strong-coupling quantum thermodynamics.

Reviewer: *The main motivation of the experiment is the test of the theory presented in Ref. 19. However, it looks like the experimental system that authors employ is too simple for this purpose and the results can be explained well by simpler and well established theories. The physical system of a qubit interacting with a harmonic oscillator realized in experiment is a closed system with no significant dissipation. Initial state of this system is close to a pure state because the initial state of the motional mode is close to the vacuum state. In this case, a traditional dressed state approach looks sufficient to explain the shift of the energy levels, as the authors themselves pointed out at the end of the supplemental information, and the concepts of “open system Ansatz of minimal dissipation” or “direct energy renormalization” are redundant and unnecessary.*

Our Response: We thank the reviewer for this detailed feedback. However, we want to clarify the motivation and contributions of our work. In the following, we address the suitability of our experimental setup, the necessity of the open-system approach, and the rationale for using the minimal-dissipation framework. Our experimental setup, though deliberately simple, is specifically tailored for testing the open-system approach proposed in Ref. 19. This controlled configuration enables us to operate in a strong coupling regime where significant dissipation effects become observable, particularly because the total system ($S + E$) is well isolated from external environments. While the overall setup is indeed well-isolated, our measurements reveal dissipative effects directly within the strongly coupled spin-mode interaction, which are essential for observing energy renormalisation in an open-system context. To clarify this, we have added further explanation in the revised manuscript and discuss the following two key points:

1. **Suitability of the Open-System Framework:** While traditional theoretical treatments, such as the dressed state and perturbation theory approach, allow for insights into time-average effects, such approaches fail to capture time-dependent shifts. The unique dissipative effects we observe in S are due to the correlations that build up between S and E . Consequently, these time-dependent shifts are only rigorously accessible through an open-system framework, which describes the dynamics of the spin system by tracing the total spin-mode state (which is close to a pure state) over the environmental mode degree of freedom, leading to a (locally) mixed state. We note that this reduced dynamics of the spin system is highly non-Markovian. While the dressed state formalism provides a complete view of the total system dynamics, it fundamentally fails to predict energy shifts in the two-level system (S) alone without focusing exclusively on the spin degrees of freedom. This perspective is clarified in the revised manuscript, where we emphasise that our results showcase the open-system approach’s unique ability to reveal effects that simpler, established methods average out.
2. **Relevance of the Minimal Dissipation Ansatz:** The minimal dissipation framework introduced in Ref. 19 is a novel and debated concept in quantum thermodynamics, providing a structured approach for defining energy renormalisation and offering measurable

predictions, in particular, for strongly coupled systems. It uniquely predicts observable time-dependent shifts in energy of the system – phenomena doubted by some in the community and unattainable with simpler theories. Our work directly addresses whether such shifts truly exist, a key point of debate in quantum thermodynamics, and provides experimental evidence to resolve this question. We see this as essential in moving the field forward. Further, we agree with the reviewer that we work in an overall closed quantum system of spin S and mode E degree of freedom. However, since S is strongly coupled to E , it shows significant open system dynamics even in a near-pure state. By combining the minimal dissipation framework with state-of-the-art techniques like Ramsey sequences, we achieved high signal-to-noise ratio measurements that connect correlation build up with (i) observed energy shifts and (ii) dissipative effects. These results validate the framework and highlight its potential to bridge theory and experimental observability for time-dependent phenomena. We see our perspectives relevant for underscoring the importance of the minimal dissipation Ansatz in advancing (strong-coupling) quantum thermodynamics.

In summary, our experimental results go beyond traditional methods by directly validating the minimal dissipation Ansatz, opening a transformative path for understanding time-dependent energy renormalisation in open quantum systems. We trust that these clarifications, together with our substantially revised manuscript, underscore the significant contributions and broad implications of our study. We show all changes in the manuscript text file with colour highlighting.

Reviewer: *In my opinion, authors failed to establish a clear connection between the theoretical hypothesis they intended to test and the experiment they have actually performed. Therefore I cannot recommend publication of the manuscript in the present form.*

Our Response: In response to the concerns regarding the connection between the theoretical hypothesis and our experimental work, we ask the reviewer to reconsider our revised manuscript. We summarize our arguments in the following three key points:

1. We have revised our manuscript to clearly outline the connection between the theoretical framework of minimal dissipation Ansatz and our experimental observations, addressing the concerns raised.
2. The solid black lines in our figures, alongside the data points, provide a direct and visual comparison between our experimental results and the predictions of the minimal dissipation Ansatz. This alignment underscores the accuracy of the theory in capturing time-dependent energy shifts under strong coupling conditions – effects that dressed states and traditional Lamb shift predictions, by design, are incapable of predicting.
3. Our observation of time-dependent shifts highlights the minimal dissipation Ansatz as a distinctly promising approach, as no other framework currently offers testable predictions for such dynamics occurring in nature. This finding validates the framework and provides a strong basis for future studies, including applications in more complex environments, such as structured baths, where these shifts are expected to be similarly relevant. Our experimental platform can be extended in size and dimension, ion by ion, to further test the validity of the Ansatz.

We are grateful for the reviewers' and editor's constructive comments, which, we hope, have significantly helped us to enhance the clarity and impact of our presentation. We look forward to further consideration and the opportunity to contribute to a hopefully ongoing dialogue.

Point-to-point response to Reviewer #2:

Reviewer: *The authors address the complex and challenging problem of understanding how strong coupling and memory effects influence the energy levels of open quantum systems. In their*

work, they demonstrate these effects by probing the transition frequency of an open two-level system within the Jaynes-Cummings model, experimentally realized through Ramsey interferometry in a single trapped $^{25}\text{Mg}^+$ ion. Their measurements, conducted while the system is coupled to a single-mode environment, reveal a time-dependent shift in the system's energy levels, reaching up to 15% of the bare system frequency. This shift, accurately predicted using an open system approach with minimal dissipation, arises solely from ultra-strong system-mode interactions and the buildup of correlations. Time-averaged measurements align with the dispersive Lamb shift predictions and match dressed-state energies, indicating that the observed shift represents a generalized Lamb shift applicable across all coupling and detuning regimes. The authors provide direct evidence of dynamic energy level renormalization in strongly coupled open quantum systems, even though the total system-environment Hamiltonian remains static. This underscores the critical role of memory effects in shaping the energy landscape of the reduced system. Their results offer deeper insights into Hamiltonian renormalization, which is vital for strong-coupling quantum thermodynamics and essential for advancements across all quantum platforms. The manuscript is well written and very clearly presented. The predictions are strongly supported by the experimental data, which are of quite high quality. The paper will definitely be interesting to a ample range of scholars, including those working in the field of finite-time quantum thermodynamics and strong-coupling quantum thermodynamics. I support the manuscript for publication in light of the exemplary clarity and quality of the results being reported.

Our Response: We thank the reviewer for their supportive and thoughtful comments on our manuscript, and we are pleased that the clarity, quality, and broader relevance of our findings were well-received. Indeed, we also believe that our findings have practical significance to test the prospects for emerging applications, such as quantum batteries and quantum thermodynamic engines, where finite-time and strong-coupling dynamics play a central role.

Lastly, in response to Reviewer 1's feedback, we have made several updates to the manuscript to clarify the connection between the theoretical framework and experimental findings. These revisions emphasise the unique predictive capability of the minimal dissipation Ansatz, highlight the necessity of the open-system perspective for capturing time-dependent shifts, and illustrate the advantages of our well-controlled experimental setup for isolating these effects. All changes are highlighted in the manuscript.